# Crosstalk between Prostate Cancer Cells and Tumor-Associated Fibroblasts Enhances the Malignancy by Inhibiting the Tumor Suppressor PLZF

**DOI:** 10.3390/cancers12051083

**Published:** 2020-04-27

**Authors:** Kum Hee Noh, Ae Jin Jeong, Haeri Lee, Song-Hee Lee, Eunhee Yi, Pahn-Shick Chang, Cheol Kwak, Sang-Kyu Ye

**Affiliations:** 1Department of Pharmacology and Biomedical Sciences, Seoul National University College of Medicine, Seoul 03080, Korea; nkhys1209@snu.ac.kr (K.H.N.); lovej89@snu.ac.kr (A.J.J.); hrlee519@snu.ac.kr (H.L.); 24happy92@snu.ac.kr (S.-H.L.); Eunhee.Yi@jax.org (E.Y.); 2Biomedical Science Project (BK21PLUS), Seoul National University College of Medicine, Seoul 03080, Korea; 3Department of Agricultural Biotechnology, Seoul National University, Seoul 08826, Korea; pschang@snu.ac.kr; 4Center for Food and Bioconvergence, Seoul National University, Seoul 08826, Korea; 5Research Institute of Agriculture and Life Sciences, Seoul National University, Seoul 08826, Korea; 6Department of Urology, Seoul National University Hospital, Seoul 03080, Korea; mdrafael@snu.ac.kr; 7Department of Urology, Seoul National University Hospital, Seoul National University College of Medicine, Seoul 03080, Korea; 8Ischemic/Hypoxic Disease Institute, Seoul National University College of Medicine, Seoul 03080, Korea; 9Neuro-Immune Information Storage Network Research Center, Seoul National University College of Medicine, Seoul 03080, Korea

**Keywords:** promyelocytic leukemia zinc finger, STAT3, prostate cancer, tumor-associated fibroblast, tumor-suppressor gene

## Abstract

Although prostate cancer is clinically manageable during the early stages of progression, metastatic progression severely compromises the prognosis and leads to mortality. Constitutive activation of STAT3 has been connected to prostate cancer malignancy, and abolishing the STAT3 activity may diminish tumor growth and metastasis. However, its suppressor genes and pathways have not been well established. In this study, we show that promyelocytic leukemia zinc finger (PLZF) has a putative tumor-suppressor function in prostate cancer by inhibiting phosphorylation of STAT3. Compared with a benign prostate, high-grade prostate cancer patient tissue was negatively correlated with PLZF expression. PLZF depletion accelerated proliferation and survival, migration, and invasion in human prostate cancer cells. Mechanistically, we demonstrated a novel role of PLZF as the transcriptional regulator of the tyrosine phosphatase SHP-1 that inhibits the oncogenic JAKs–STAT3 pathway. These results suggest that the collapse of PLZF expression by the CCL3 derived from fibroblasts accelerates the cell migration and invasion properties of prostate cancer cells. Our results suggest that increasing PLZF could be an attractive strategy for suppressing prostate cancer metastasis as well as for tumor growth.

## 1. Introduction

Prostate cancer is the most frequently diagnosed malignancies and the sixth leading cause of cancer-related death in men worldwide [1]. In general, patients with organ-confined prostate cancer are treated successfully. However, uncontrolled metastasis indicates disease relapse, which is virtually incurable and results in considerable disease mortality [2,3,4]. In addition, although much research has been done to determine the cause of prostate cancer metastasis, our understanding is still incomplete. For this reason, a better understanding of prostate cancer metastasis would lead to new approaches and targets for preventing or treating metastasis [5].

Tumors can be considered as complex organs composed of tumor cells and various non-malignant stromal cells that form the tumor microenvironment (TME). These stromal cells include endothelial cells, pericytes, immune inflammatory cells, and fibroblasts, all of which are genetically stable and typically not malignant [6,7,8]. However, they are affected by interactions with tumor cells via soluble factors and modify tumor cells to favor tumor development, tumor growth, and invasion [8,9,10]. Among them, fibroblasts are crucial regulators of prostate tumor metastatic progression [11,12]. Several researches have investigated that the normal fibroblasts have a role in maintaining epithelial homeostasis by inhibiting the proliferation and carcinogenic potential of the adjacent epithelium; but, following neoplastic transformation of the epithelium, tumor-associated fibroblasts promote tumor growth and remodels the extracellular matrix (ECM). That is, normal fibroblasts can be trained by carcinoma cells to become tumor-associated fibroblasts [13,14,15].

The mechanism by which local prostate cancer progresses into a lethal disease is unclear. One predictable mechanism is the loss or alteration of tumor-suppressor genes. Promyelocytic leukemia zinc finger (PLZF), also known as Zinc finger and BTB domain-containing 16 (ZBTB16), binds to specific DNA sequences with its carboxy-terminal zinc finger domain and protein–protein interaction [16,17]. PLZF is involved in diverse cellular signaling, particularly in stem cells self-renewal or differentiation, hematopoiesis, and immune regulation [18,19,20,21,22]. Lately, PLZF was reported to potentially play a tumor-suppressor role, which reduces cell growth and survival in numerous solid tumors, including melanoma, malignant mesothelioma, and non-small cell lung cancer cells [23,24,25,26]. Besides, recent studies also suggested that PLZF is implicated in prostate cancer as a tumor-suppressor protein. In prostate cancer, PLZF expression is reduced or lost in high grade tumor and castration-resistant prostate cancer (CRPC) [17,27]. PLZF inhibits prostate cancer cell growth through its inhibitory effects on AR, AKT, mTOR, and MAPK signaling [28,29]. Research of PLZF is actively underway as it has a high potential as a target for prostate cancer, but the relationship between prostate tumor microenvironment and PLZF is not yet well known. Therefore, we aim to clarify the role of PLZF in the association between prostate cancer and the tumor microenvironment. Restoring PLZF expression or reactivation also could be a novel strategy for prostate cancer therapy.

Surprisingly, we discovered that tumor-associated fibroblasts inhibited PLZF expression. In this study, we hypothesized that PLZF is suppressed by tumor-associated fibroblasts, thereby promoting metastatic prostate cancer progression. As a result, we found that tumor-induced impetus stimulated fibroblasts to produce CCL3, which promoted prostate cancer growth and metastasis through the reduction of PLZF/tyrosine phosphatase SHP-1. By studying the PLZF/SHP-1/STAT3 signaling pathway involved in CCL3-induced malignancy, we may find a potential therapeutic target for prostate cancer.

## 2. Results

### 2.1. PLZF Inhibits the Activation of STAT3, Which Has an Inverse Correlation with PLZF

We first evaluated the protein levels of PLZF and Tyr705 phosphorylation STAT3 (pY-STAT3) in 40 prostate cancer patients and 10 benign patient tissues, which were categorized according to Gleason scores (GS). Immunohistochemistry (IHC) assays showed that the levels of pY-STAT3 increased with the progression of prostate cancer, as opposed to the decrease in PLZF levels (Figure 1A, Appendix A). Pearson’s correlation analyses revealed a significant negative correlation between PLZF and pY-STAT3 levels (Figure 1B). The Prostate Specific Antigen (PSA) from prostate cancer patients also has been shown to increase with Gleason scores (Appendix A, Appendix A). The pY-STAT3 protein levels are correlated, but the PLZF protein levels are inversely correlated with PSA (Figure 1C). Moreover, prostate cancer patients from The Cancer Genome Atlas (TCGA) database were divided into PLZF- and pY-STAT3-low/high groups. Kaplan–Meier analysis indicated that patients with PLZF-low and pY-STAT3-high groups had poorer recurrence-free survival than those with PLZF-high and pY-STAT3-low expression (Figure 1D). In addition, the higher-PLZF group also showed higher survival rates than the lower groups in the CRPC patient group with the poorest prediction for prostate cancer (Appendix A). We compared the PLZF mRNA expression between a normal, primary tumor, and metastatic tumor using the TCGA database, and found that PLZF expression was more decreased in metastatic tumors (Figure 1E). We examined PLZF, pY-STAT3, and STAT3 protein/mRNA levels by Western blotting/RT-PCR in prostate cancer cell lines (Figure 1F, Appendix A). Interestingly, PLZF was not affected by exogenous constitutive activation (CA) of STAT3, but it was discovered that pY-STAT3 was decreased by PLZF. Based on the gene-silencing efficiency (Appendix A), we chose the #5 siRNA directed against Human PLZF (SI03090346). It was also confirmed that knockdown of PLZF increased the phosphorylation of STAT3 (Figure 1G). Reducing pY-STAT3, a well-known oncogene in prostate cancer, suggests that PLZF is likely involved in progression of prostate cancer. Therefore, we hypothesized that PLZF acts as the tumor-suppressor gene to improve the poor prognosis in prostate cancer patients.

### 2.2. PLZF Induces the Cell Cycle Arrest and Apoptosis Effects by Suppression of STAT3 Signaling

To investigate the tumor-suppressing role of PLZF in prostate cancer, we overexpressed PLZF in DU145 cells. Overexpression of PLZF resulted in significantly reduced proliferating cell nuclear antigen (PCNA) protein expression, and inhibited cell growth. In contrast, knockdown of endogenous PLZF increased cell viability in LNCaP cells (Figure 2A,B, Appendix A). Moreover, to explain that PLZF functions as a tumor suppressor, cell cycle distribution was detected. Cell cycle examined by flow cytometry analysis revealed that 10.825% more cells increased in the sub-G1 phase proportion and 14.75% more cells accumulated in the G0/G1 phase compartment with PLZF-overexpressed DU145 cells (*n* = 3) (Figure 2C). To confirm the molecular mechanism of PLZF, the expression levels of the cell cycle arrest regulators, including c-MYC, cyclin D1, cyclin D3, CDK4, p21, and p27, were tested. As a result, the G0/G1 phase arrest is confirmed by PLZF (Figure 2D,E). In addition to promoting cell cycle arrest, PLZF triggered prostate cancer cell apoptosis, with effectively increasing the apoptosis proportion in the Annexin V-FITC/PI staining assay. Compared to the controls, an increase in the percentage of early and late apoptotic cells was observed in PLZF-overexpressed DU145 (early, from 7.43% to 14.83%; late, from 3.47% to 10.69%; Figure 2F). The mRNA/protein levels of the BCL-2 family apoptotic markers were inhibited in PLZF-overexpressed cells, but they were increased in PLZF-knockdown cells (Figure 2G,H). As a result, these findings indicated that the increase in PLZF expression in prostate cancer cells induced cell cycle arrest and apoptosis.

### 2.3. PLZF Ablation in Prostate Cancer Promotes Cell Migration and Invasion via Activation of STAT3

We next evaluated the migration and invasion ability by wound healing assays, Transwell cell migration assays, and Matrigel invasion assays. Quantitative analysis of the wound healing assay revealed that PLZF-transfected cells delayed the closure of the wound gap as well as delayed the STAT3-knockdown in DU145 cells. (Figure 3A, Appendix A). Consistently, a similar effect of PLZF was observed in the Transwell cell migration assays. As with the siSTAT3-transfected cells, the number of migrated cells with ectopically expressed PLZF showed an apparent reduction compared to the control cells. In contrast, knockdown of endogenous PLZF in LNCaP cells by siRNA increased the cell migration ability. The Matrigel invasion assays demonstrated that the cell invasion ability is significantly attenuated by PLZF-overexpressed DU145 cells but is increased by PLZF-knockdown LNCaP cells (Figure 3B). The mRNA/protein levels of the epithelial marker (E-cadherin) were increased in PLZF-overexpressed cells but inhibited in PLZF-knockdown cells. In mesenchymal markers (N-cadherin, Vimentin and Fibronectin), the opposite was observed (Figure 3C,D). Furthermore, consistent with an epithelial–mesenchymal transition (EMT) phenotype, DU145 cells exhibited diminished ECM degradation as evidenced by decreased levels of MMP2 and MMP9 expression (Appendix A). The data presented above suggested PLZF could suppress prostate cancer migratory and invasive abilities by preventing the EMT process.

### 2.4. Tyrosine Phosphatase SHP-1 Is a Direct Target of PLZF and Inhibits the Tyrosine Phosphorylation of JAKs–STAT3 Signaling

To interrogate the molecular mechanism of PLZF, which has been demonstrated as a tumor-suppressor gene in human prostate cancer cells, and whether it is associated with the oncogenes JAK1, JAK2, and JAK3 as well as the TYK2(JAKs)–STAT signaling molecules, we ectopically transfected PLZF in DU145 cells. As a result, PLZF inhibits the phosphorylation of the JAKs–STAT3 signaling pathway molecules but not in STAT1 and STAT5. Especially, tyrosine phosphorylation of the JAKs–STAT3 proteins was inhibited in PLZF-overexpressed DU145 cells. In contrast, serine phosphorylation and the total JAKs–STAT proteins remained unchanged (Figure 4A). The mRNA levels of STAT3 were also not significantly changed in PLZF-overexpressed cells (Appendix A). Besides, pY-STAT3 and total STAT3 proteins did not bind with PLZF (Appendix A). If only tyrosine, not serine, was reduced by PLZF during phosphorylation of STAT3, the status of the phosphorylation would be questioned when phosphorylated STAT3 was translocated into the nucleus. We performed cytosol–nuclear fractionation with PLZF-overexpressed DU145 cells. The nuclear and cytoplasmic proteins were dissociated, and the Western blot showed that exogenous overexpression of PLZF significantly decreased only tyrosine phosphorylation of STAT3, not serine phosphorylation of STAT3 in both the cytosol and nucleus compared to the control (Appendix A). Since PLZF suppresses tyrosine phosphorylation only, it is assumed that PLZF can dephosphorylate JAKs–STAT3 through tyrosine phosphatase. To identify the tyrosine phosphatases related to STAT3, the mRNA/protein levels of protein tyrosine phosphatase SHP-1 were increased in PLZF-overexpressed cells, but they were inhibited in PLZF-knockdown cells (Figure 4B,C). Based on the gene-silencing efficiency (Appendix A), we chose the #10 siRNA directed against Human SHP-1 (SI04436831). PLZF overexpression significantly induced SHP-1 expression, whereas tyrosine-phosphorylation of JAKs–STAT3 was abolished by ectopic-PLZF despite SHP-1 ablation (Figure 4D).

To determine whether the SHP-1 promoter region contributes to transcriptional regulation of the PLZF gene, we constructed a pGL3-luciferase reporter plasmid containing the SHP-1 promoter fragment. In prostate cancer cell lines, SHP-1 promoter activity was increased by PLZF overexpression, while the activity was diminished by PLZF knockdown (Figure 4E). The IHC quantification of SHP-1 protein expression demonstrated that the protein levels are lower in high GS (Figure 4F). Pearson’s correlation analyses revealed a prominently positive correlation between SHP-1 and PLZF levels (Figure 4G). The SHP-1 protein levels of prostate cancer patient tissues are inversely correlated in PSA (Figure 4H). Moreover, prostate cancer patients from the TCGA database were divided into the SHP-1 high (*n* = 71)/low (*n* = 70) expression groups. Kaplan–Meier analysis indicated that patients of low SHP-1 levels had poorer recurrence-free survival than those with high SHP-1 levels (Figure 4I). These results indicate that PLZF suppressed the tyrosine phosphorylation of JAKs–STAT3 by promoting SHP-1 transcription, acting as a tumor-suppressor gene.

### 2.5. PLZF Is Reduced in Prostate Cancer Due to the Prostate Tumor-Associated Fibroblasts

Fibroblasts are one of the most abundant cell types in the prostate stroma. Fibroblasts synthesize many components of ECM (collagens I, II, and V, and fibronectin) and are responsible for the remodeling of ECM by secreting matrix metalloproteinases (MMPs). Fibroblasts also secrete growth factors and cytokines to regulate differentiation and homeostasis of adjacent epithelium. In some physiological (inflammation) or pathological (cancer) conditions, fibroblasts have an activated phenotype due to stimulation by growth factors or cytokines [30]. In prostate cancer progressions, interactions with various tumor microenvironment cells, including cancer cells and fibroblasts, are known to play an important role [8,11,12]. In this study, we hypothesized that prostate tumor-associated fibroblasts suppress PLZF expression and enhance malignant behaviors. To examine the cells affecting cancer cells among various stromal cells in the tumor microenvironment, LNCaP cells were co-cultured with lymphocytes (Jurkat), NK cells (NK92), monocytes (THP-1), and fibroblasts. Interestingly, only the LNCaP sample co-cultured with the fibroblast suppressed PLZF levels (Appendix A). To determine the influence of fibroblasts on prostate cancer, we adopted a co-culturing system using Transwell chambers (Figure 5A). Surprisingly, we found that both protein and RNA expression of PLZF were more reduced by direct co-culturing of fibroblast and cancer cells than fibroblast CM (FCM). The direct co-culturing systems enable the sharing of medium and soluble factors throughout the entire culture well. These results indicated that co-culturing with fibroblast was expected to promote the malignancy of prostate cancer cells (Figure 5B,C). In addition, both protein and RNA expression of PLZF were confirmed with the same results in PLZF-overexpressed DU145 cells as in LNCaP cells (Appendix A). These results indicated that co-culturing with fibroblast was expected to promote the malignancy of prostate cancer cells. To understand the functional role of FCM and direct co-culturing in prostate cancer, we examined cell viability, migration, and invasion assays. LNCaP cells viability was further increased by the co-cultured CM (=Co-CM) than FCM through the CCK assay (Figure 5D). Next, Transwell cell migration and Matrigel invasion assays were examined, and Co-CM significantly promoted cell migration and invasion in LNCaP cells rather than FCM (Figure 5E, Appendix A). The mRNA/protein levels of the mesenchymal markers were more increased in the LNCaP cells treated with Co-CM, whereas E-cadherin was more decreased than FCM (Figure 5F,G). To confirm that normal fibroblasts increased the malignancies of cancer cells, non-normal cells, the normal prostatic epithelial cell line (PNT2), and LNCaP cells were co-cultured with fibroblasts. In PLZF-overexpressed PNT2 cells, the PLZF levels were not significantly affected by co-culturing with fibroblasts; however, PLZF levels were substantially inhibited by co-culturing with fibroblasts in LNCaP cells (Figure 5H). Taken together, these findings further support the functional significance of fibroblasts in the tumor microenvironment, in that they remarkably promote the malignant behavior of prostate cancer cells.

### 2.6. Loss of CCL3 Inhibits the Fibroblast-Induced Prostate Cancer Cells Migration and Invasion

To confirm whether the fibroblast-derived factors in CM is a polypeptide, Co-CM was treated or heated with mixture of CM and fresh medium. Consequently, the fibroblast-inducing effect was practically abolished by heat inactivation (Appendix A). To explore the soluble mediators released by the fibroblast and fibroblast co-cultured with prostate cancer cells, we analyzed by using protein arrays the levels of multiple cytokines, chemokines, and growth factors (Figure 6A, Appendix A). These candidates that were more enriched in co-cultured CM (Co-CM) than prostate cancer cell CM (CTL CM) and fibroblast CM (FCM) were identified: C–C motif chemokine ligand 3/4 (CCL3/4), serpinE1, and urokinase-type plasminogen activator receptor (uPAR). To determine the cytokine responsible for PLZF suppression, each of the four candidates was knocked down in fibroblast and cancer cells using siRNAs. Knockdown of CCL3 or uPAR, but not CCL4 or serpinE1, attenuated the PLZF protein and mRNA expression suppression by FCM and Co-CM (Appendix A). Based on the gene-silencing results, in order to evaluate the increase in the identified cytokines, fibroblasts were co-cultured with the normal prostate cells (PNT2) or the prostate cancer cells (LNCaP, DU145). In contrast to no effect on CCL4 and SerpinE1 mRNA expression, fibroblasts visibly induced CCL3 and uPAR mRNA production by co-culturing with prostate cancer cells (Figure 6B). To confirm the PLZF-suppressing effects of the two cytokines (CCL3, uPAR), neutralizing antibodies were administered to FCM and Co-CM. Only anti-CCL3 alone abolished the PLZF-inhibitory effect (Figure 6C). We next performed the ELISAs; CCL3 is secreted at higher levels in Co-CM than CTL and fibroblasts alone (Figure 6D). In this study, PLZF acts as a transcription factor of the SHP-1 gene (Figure 4). Therefore, to investigate whether CCL3 inhibits PLZF-mediated transcription of the SHP-1 gene, prostate cancer cells were treated with a recombinant CCL3. As a result, SHP-1-driven transcription was more inhibited with Co-CM than FCM by CCL3 (Figure 6E, Appendix A). These results suggest that if fibroblasts present near cancer cells, prostate tumor-associated fibroblasts are expected to produce CCL3 more abundantly than normal fibroblast cells. Moreover, by neutralizing CCL3, cell migration and invasion abilities were further abolished by Co-CM, more so than FCM (Figure 6F). As a result of the confirmed mRNA/protein levels reinforcing the EMT assays, the alteration in EMT markers expression was abolished by the antibody of CCL3 (Appendix A). Collectively, these results indicate that CCL3 secreted by fibroblasts co-cultured with prostate cancer cells inhibits tumor suppressor PLZF expression in prostate cancer.

## 3. Discussion

In this study, we discovered a new role for fibroblast in the prostate-tumor microenvironment to inhibit the tumor suppressor PLZF. While numerous reports have described the coevolution of stroma and prostate cancer cells in genotypic and phenotypic characters, some studies have elaborated on the contribution of tumor-associated fibroblasts to overall changes in gene and protein expression in prostate cancer microenvironments [31,32]. We identified for the first time a mechanism by which fibroblast-derived CCL3 suppresses PLZF expression, which promotes the synthesis of SHP-1 at the transcriptional levels, and then activates the JAKs–STAT3 pathway. PLZF was also significantly reduced in prostate cancer patient tissues, and low PLZF expression was found to correlate with GS and recurrence-free survival. Accordingly, down-regulation of PLZF activities can promote metastasis progression through improved growth and invasion of prostate cancer.

Cancer cells make use of the tremendous plastic nature of stromal cells, such as fibroblasts and macrophages, and produce multiple signals that generate a tumor-promoting microenvironment [33,34,35]. Previous studies have indicated that co-culture or tumor cell CM could activate stromal fibroblasts, and tissue around fibroblasts were suggested as precursors for tumor-associated fibroblasts, activated by tumor cells [34,35,36]. In addition, fibroblasts are constantly exposed to different stimuli in the tumor microenvironment, facilitating unique features, such as excessive and specific secretion and the ECM remodeling phenotype [37,38]. Activated fibroblasts, i.e., cancer-associated fibroblasts (CAF), are involved in tumor invasiveness and metastasis by expressing a series of proteins that are not expressed by normal fibroblasts, such as α-smooth muscle actin (α-SMA) [39,40]. Moreover, contact between cancer cells and normal fibroblasts can facilitate the CAF phenotype in breast cancer through Notch signaling [41]. Various inflammatory modulators can promote CAF activation by interleukin-1 (IL-1), which acts through NF-κB and IL-6, which mainly acts on signal transducer and activator of transcription (STAT) transcription factors [15]. Crosstalk and positive feedback involved in Janus kinase (JAK)–STAT signaling transduction, the contractile cytoskeleton, and alterations in histone acetylation further promote CAF activation [42,43]. It has been reported that tumor-associated fibroblasts promote tumor growth by directly stimulating tumor cell proliferation, play an important role in wound healing, and mediate other aspects of complex processes such as extracellular matrix remodeling [10,15,44]. Moreover, tumor-associated fibroblasts contribute to tumor progression by providing cancer cells with growth factors and pro-inflammatory tumor-promoting microenvironments [15,45,46].

CCL3, which belongs to the C–C chemokine family, is produced by fibroblast, macrophage, T cell, monocytes, and epithelial cells [47,48]. Previously, several evidence models suggested that fibroblast-derived CCL3 could activate tumor progression. Deficiency of the CCL3/CCR5 system resulted in a remarkably decreased tumor formation and lung metastasis. CCL3/CCR5 accumulation of tumor-associated fibroblasts also have been reported to be important in colitis-related carcinogenesis and oral tumor formation [46,49,50,51]. However, the fundamental mechanisms of CCL3 in the tumor microenvironment and the signaling pathways are not well known. We demonstrated that the expression of the marker PLZF was significantly increased in the absence of CCL3.

We have identified how various oncogenes related to prostate tumor survival are altered by PLZF. It was surprisingly found that only tyrosine phosphorylation of JAKs–STAT3 was reduced, so we focused on tyrosine phosphatase SHP-1 related to STAT3. The cytoplasmic protein tyrosine phosphatase (PTP), characterized by containing two Src homology 2 (SH2) N-terminus, a single catalytic domain, and a C-terminal PTP domain, is referred to as SHP-1. The expression of SHP-1 in prostate cancer is inversely correlated with tumor stage and malignancy, as well as with biochemical recurrence after prostatectomy [51,52,53,54]. In addition, SHP-1 is known to have a tumor-suppressor role in various cancers due to the SHP-1-negative regulation of JAKs–STAT activation via growth factors and cytokines [54,55,56,57]. We found that SHP-1 transcription was promoted by PLZF, which reduced tyrosine phosphorylation of JAK2, TYK2, and STAT3. Collectively, these results suggest that PLZF can function with SHP-1 to reduce prostate cancer progression.

As a result, our current findings suggest a crucial mechanism involving positive feedback between fibroblast and cancer cells, and this mechanism is essential for the progression and metastasis of prostate cancer. To our knowledge, we have identified for the first that the CCL3/PLZF/SHP-1/pY-STAT3 signaling cascades may be a target for combating prostate cancer cells. In conclusion, considering its interactions with several key signaling pathways in the prostate tumor microenvironment, understanding PLZF may help develop new biomarkers and therapeutic strategies for prostate cancer.

## 4. Materials and Methods

### 4.1. Cell Culture and Culture Conditions

The human prostate cancer (LNCaP) cell line was purchased from the Korean Cell Line Bank (Seoul, Korea). Another human prostate cancer (DU145) and normal human dermal fibroblast cell lines were kindly provided by the German Collection of Microorganisms (DSMZ, Braunschweig, Germany) and Modern Cell & Tissue Technologies (Seoul, Korea). The normal prostate epithelial PNT2, Jurkat-T, NK92 and THP-1 were purchased from American Type Culture Collection (Manassas, VA). DU145 cells was maintained in Dulbecco’s Modified Eagles medium (Capricorn scientific, Ebsdorfergrund, Germany). LNCaP, PNT2, Jurkat-T, NK-92, THP-1 and normal human dermal fibroblast cells were maintained in RPMI-1640 medium (Life Technologies, Carlsbad, CA, USA). All the cells were maintained in the medium supplemented with 10% heat-inactivated fetal bovine serum (FBS, Life Technologies, Carlsbad, CA, USA) and 1% penicillin/streptomycin solution (Life Technologies, Carlsbad, CA, USA) at 37 °C with 5% CO_2_ in a humidified incubator.

### 4.2. Immunohistochemistry Staining and Scoring

Briefly, the arrays were rehydrated and autoclaved at 121 °C for 10 min in a citrate buffer (Fisher Scientific, Rockford, IL, USA) to retrieve the antigens. They were incubated with 3% H_2_O_2_ for 10 min and with 10% bovine serum for 1 h to block non-specific signals. Sections were incubated overnight at 4 °C with antibodies against PLZF (1:200; abCam, Cambridge, MA, USA), pY-STAT3 (1:50; cell signaling, Danvers, MA, USA), and SHP-1 (1:200; abCam). Immune complexes were visualized using the Vectastatin ABC kit (Vector Laboratories, Burlingame, CA, USA) and the DAB detection kit (Dako, Carpinteria, CA, USA). Finally, all sections were counterstained with hematoxylin. Positive staining cells were counted on 5 randomly chosen visual fields at 400× magnification by means of ImageJ (National Institutes of Health, Bethesda, MD, USA).

### 4.3. Human Prostate Cancer Patients Tissues

Human prostate cancer tissue arrays were purchase from SuperBioChips Lab (Seoul, Korea). The tissue array contained 40 tumor specimens, as well as 10 normal tissues adjacent to the cancer from prostate cancer patients, whose clinical data, including age, sex, TNM stage, and Gleason score, were informed by the supplier. The tumor tissues were fixed with formalin, paraffin-embedded, and sectioned by a microtome to a 4-μm thickness and put on Superfrost plus slides.

### 4.4. Cell Migration and Invasion Assays

Transwell assay was used to evaluate the migration and invasion abilities of the prostate cancer cells. Twenty-four-well Transwell chambers with an 8 µm pore size polycarbonate membrane (Corning Inc., Corning, NY, USA) were used in these assays. For the Transwell assays, the PLZF expression vector and PLZF siRNA were transfected into DU145 or LNCaP cells and resuspended at a concentration of 5 × 10^4^ cells in the top well with 200 µL serum-free RPMI 1640 medium. The lower chamber was filled with a 700 µL mixture of fresh and CMs (1:1) with 10% fetal bovine serum or neutralizing antibody CCL3 (R&D System, Minneapolis, MN, USA) were added as a chemoattractant. For cell migration assay, DU145 or LNCaP cells in a serum-free medium were seeded into the top chamber and incubated at 37 °C for 24 h. For cell invasion assay, the membrane was coated with Matrigel (Corning Inc., Corning, NY, USA) and incubated at 37 °C for 36 h. Cells on the top surface of the interface membrane were removed using a cotton swab. Migrated and invaded cells on the lower surface of the membrane were fixed with 10% formaldehyde, stained with a hematoxylin and eosin kit (Sysmex Corporation, Kobe, Japan), and counted under an optical microscope (100×, Nikon, Tokyo, Japan) from four random fields using ImageJ software (National Institutes of Health, Bethesda, MD, USA).

### 4.5. Luciferase Gene Report Assay

For the PLZF target reporter assay, DU145 and LNCaP cells were transiently co-transfected with 1 μg of the pGL3-SHP-1 promoter-luciferase reporter construct, pcDNA3.1-PLZF or si-PLZF, and pCMV-β-galactosidase. After 48h, cells were lysed and luciferase activity was determined using Luciferase reporter assay system (Promega, Madison, WI, USA). β-galactosidase activity in each lysate was assayed to normalize transfection efficiency. Three independent experiments were performed with triplicate samples.

### 4.6. Conditioned Media Preparation

Conditioned media (CM) for treating LNCaP cells was prepared by incubating normal human dermal fibroblast, Jurkat, NK92 and THP1 cells in 10mL dishes for 48 h with RPMI-1640 medium (Life Technologies, Carlsbad, CA, USA) containing only 1% penicillin/streptomycin solution (Life Technologies, Carlsbad, CA, USA). Supernatant was collected and centrifuged at 13,000rpm for 3 min to remove the cell debris. Cells were counted and CMs were normalized by adding media to a final volume that would represent 1 × 10^6^ cells per mL. LNCaP cells were cultured in a mixture of indicated CMs and fresh media for 24 h.

### 4.7. Conditioned Human Proteome Profiler Cytokine Arrays

Human cytokine array kits were used according to the manufacturer’s protocol (R&D System, Minneapolis, MN, USA). Although membranes were blocked for 1 h, culture supernatants from human dermal fibroblast cells were added to the membranes and incubated with cytokine arrays membranes overnight at 4 °C. Membranes were washed and incubated with detection biotinylated antibody cocktail (1:50) at room temperature for 1 h and incubated with horseradish-peroxidase-conjugated streptavidin (1:2000) at room temperature for 30 min, developed using a chemi-luminescence-type solution, and exposed to X-ray film. Spot intensities were analyzed using Adobe Photoshop.

### 4.8. ELISA Assays

Human CCL3 levels in supernatants of LNCaP cells, normal human dermal fibroblasts, and co-cultured LNCaP/fibroblasts were analyzed by commercial kits purchased from R&D Systems (Minneapolis, MN, USA), according to the manufacturer’s instructions. Positive controls were supplied in the kit.

### 4.9. Microscopy, Image Capture, and Analysis

Samples were visualized and analyzed using a Leica Aperio AT Turbo microscope and the Leopard program included with the microscope (BX53; Olympus, Tokyo, Japan), using the Leica Application Suite software. Brightness and contrast were adjusted equally in all images presented.

### 4.10. Statistical Analysis

All experiments were performed with Microsoft Excel 2016 or GraphPad Prism 5 analytical tools (GraphPad software, Inc.), and results were presented as the means ± SD from least three independent samples. Student’s unpaired two-tailed t-test for independent analysis was applied to evaluate the differences. Pearson’s correlation was used to evaluate the correlation between PLZF, pY-STAT3, and SHP-1 protein expression. Survival rate analyses were performed by drawing curves and calculating log-rank *p* test using the Kaplan–Meier method. A *p*-value less than 0.05 was considered statistically significant. * *p* < 0.05, ** *p* < 0.01, *** *p* < 0.001.

## 5. Conclusions

In conclusion, our results strongly suggest that PLZF acts as a prostate-tumor suppressor, blocking cell viability, migration, and invasion. Furthermore, prostate cancer cooperates with tumor-associated fibroblasts to collapse PLZF that occurred by CCL3 derived from the tumor-associated fibroblasts. Mechanistic studies revealed that loss of PLZF by CCL3 activates the oncogenic JAK/STAT3 pathways through reduction of SHP-1 (Figure 7). Therefore, PLZF may act as a potential therapeutic target for prostate cancer.

## Figures and Tables

**Figure 1 cancers-12-01083-f001:**
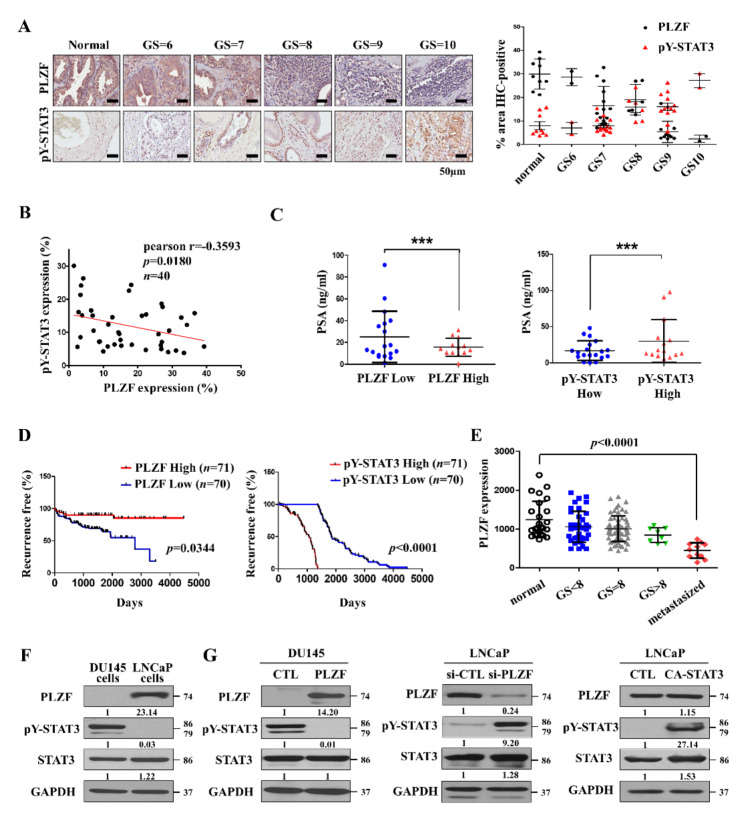
Promyelocytic leukemia zinc finger (PLZF) inhibits the activation of STAT3, which has an inverse correlation with PLZF. (**A**) IHC of PLZF, pY-STAT3 in tissue arrays of surrounding normal prostate tissues, and human prostate cancer specimens correlates with Gleason scores (GS). Quantification of PLZF-positive expression according to the GS in normal tissues adjacent to the benign tumors (*n* = 9) and malignant tumors (*n* = 40) (right). (**B**) Associations between expression of PLZF and pY-STAT3. Scatter plots showing the linear correlation determined by Pearson correlation coefficient calculation of those genes that were statistically significant. Pearson correlation coefficient r and *p*-values are given in each scatter plot. (**C**) PSA levels were measured according to the PLZF and pY-STAT3 protein levels, *n* = 40. (**D**) Kaplan–Meier recurrence-free survival analysis of prostate cancer patients according to PLZF (* *p* = 0.0344) and pY-STAT3 (* *p* < 0.0001) expression. (**E**) Quantification of PLZF mRNA expression according to the GS and metastasis in prostate cancer patients’ samples, *** *p* < 0.0001. (**F**) PLZF, pY-STAT3, STAT3, and GAPDH protein expression by Western blotting in the prostate cancer cell lines DU145 and LNCaP. GAPDH was used as a loading control. (**G**) Western blotting was performed in PLZF, CA-STAT3 plasmid, and siRNA-transfected cells. The uncropped blots and molecular weight markers of Figure 1 are shown in Appendix A

**Figure 2 cancers-12-01083-f002:**
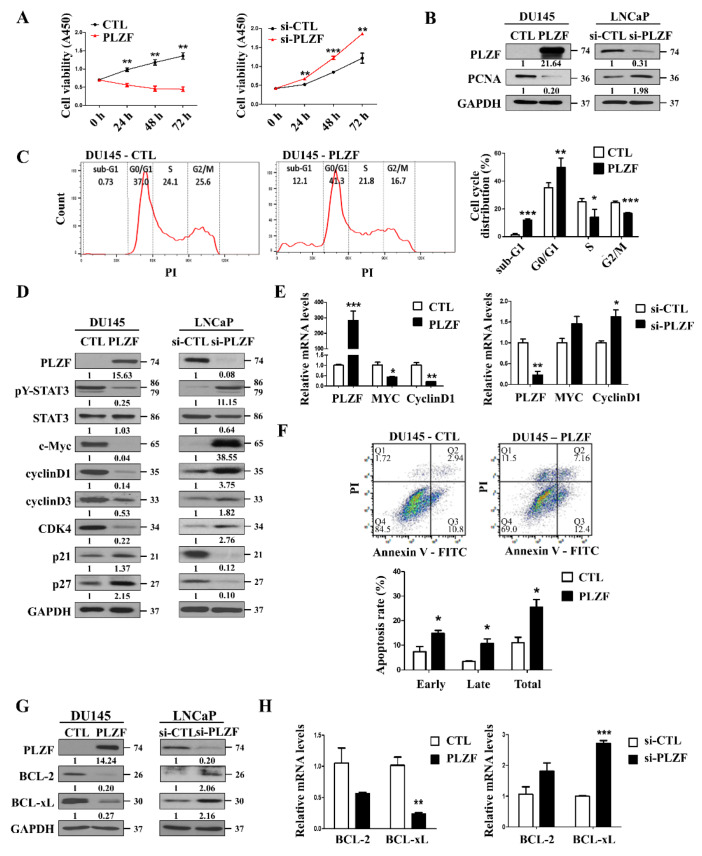
PLZF induces cell cycle arrest and apoptosis effects by suppression of STAT3 signaling. (**A**) CCK assay was performed by transfecting DU145 and LNCaP cells with plasmid and siRNA, followed by culture for 1–3 days. (**B**) Western blotting was performed in PLZF plasmid and siRNA-transfected cells. (**C**) Effect of cell cycle distribution of MOCK- and PLZF-transfected DU145 cells was detected by flow cytometry analysis. Representative histograms of cell cycle alteration. Summarized results from three independent experiments were quantified as mean ± SD (right). (**D**) Protein expression levels of indicated cell cycle regulators were detected by Western blotting. (**E**) mRNA expression levels of PLZF, MYC, and CyclinD1 were examined by qRT-PCR in DU145 and LNCaP cells transfected with PLZF plasmid and siRNA. (**F**) Apoptosis assay of PLZF plasmid and siRNA transfected DU145 and LNCaP cells was detected by Annexin V-FITC/PI staining. Representative histograms of cell cycle alteration. Summarized results from three independent experiments were quantified as mean ± SD (right). (**G**) Protein expression levels of indicated apoptosis regulators were detected by Western blotting. (**H**) mRNA expression levels of BCL2 and BCLxL were examined by qRT-PCR in DU145 and LNCaP cells transfected with PLZF plasmid and siRNA. In (A) and (B), data are presented as the mean ± SD; * *p* < 0.05, ** *p* < 0.01, *** *p* < 0.001. The uncropped blots and molecular weight markers of Figure 2 are shown in Appendix A

**Figure 3 cancers-12-01083-f003:**
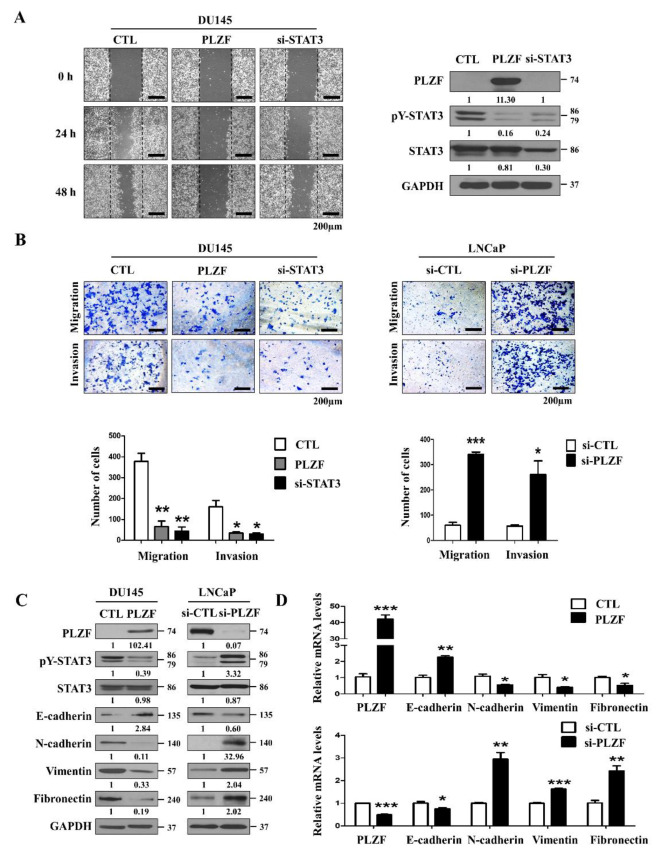
PLZF ablation in prostate cancer promotes cell migration and invasion via activation of STAT3. (**A**) Wound healing assay (left) and Western blotting (right) was conducted in PLZF plasmid and siSTAT3-transfected DU145 cells. (**B**) Transwell migration and Matrigel invasion assays were conducted in MOCK-/PLZF-/siSTAT3- and siNT-/siPLZF-transfected cells. The relative cell numbers are shown (bottom). (**C**) Protein expression of the epithelial/mesenchymal related markers were compared by Western blotting in MOCK-/PLZF-transfected cells and siNT-/siPLZF-transfected cells. (**D**) mRNA expression levels of epithelial/mesenchymal related markers were examined by qRT-PCR in DU145 and LNCaP cells transfected with PLZF plasmid and siRNA. In (A) and (B) representative images from three independent experiments were quantified as the mean ± SD; * *p* < 0.05, ** *p* < 0.01, *** *p* < 0.001. The uncropped blots and molecular weight markers of Figure 3 are shown in Appendix A

**Figure 4 cancers-12-01083-f004:**
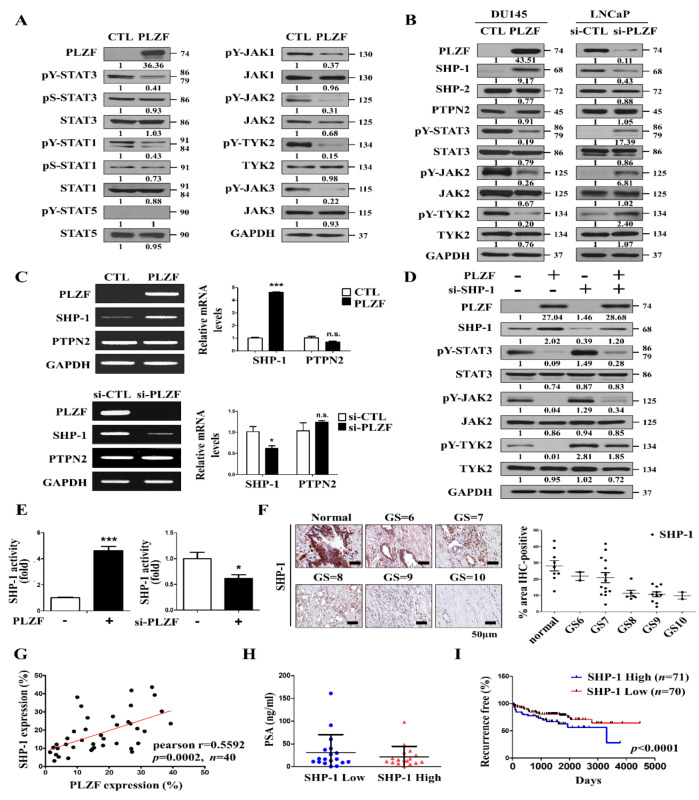
Tyrosine phosphatase SHP-1 is a direct target of PLZF and inhibits the tyrosine phosphorylation of JAKs–STAT3 signaling. (**A**) Western blotting was performed in PLZF-transfected DU145 cells. (**B**) Protein expression levels of the tyrosine phosphatase-related markers STAT3, JAK2, and TYK2 were detected by Western blotting. (**C**) mRNA expression levels of tyrosine phosphatases were examined by RT-PCR and qRT-PCR in DU145 and LNCaP cells. (**D**) Western blotting of SHP-1, STAT3, JAK2, and TYK2 in cells that were co-transfected with PLZF and si-SHP-1. (**E**) SHP-1 reporter assay in DU145 and LNCaP cells transfected with PLZF plasmid and siRNA. (**F**) IHC of SHP-1 in tissue arrays of surrounding normal prostate tissues and human prostate cancer specimens correlates with GS. Quantification of SHP-1-positive expression according to the GS scores in benign (*n* = 9) and malignant tumors (*n* = 40) (right). (**G**) Associations between SHP-1 and PLZF. Scatter plots showing the linear correlation determined by Pearson correlation coefficient calculation of those genes that were statistically significant. Pearson correlation coefficient r and p-values are given in each scatter plot. (**H**) PSA levels were measured according to the SHP-1 protein levels, *n* = 40. (**I**) Kaplan–Meier recurrence-free survival analysis of prostate cancer patients according to SHP-1 (* *p* < 0.0001) expression. In C, E, and F data are presented as the mean ± SD; * *p* < 0.05, *** *p* < 0.001. The uncropped blots and molecular weight markers of Figure 4 are shown in Appendix A

**Figure 5 cancers-12-01083-f005:**
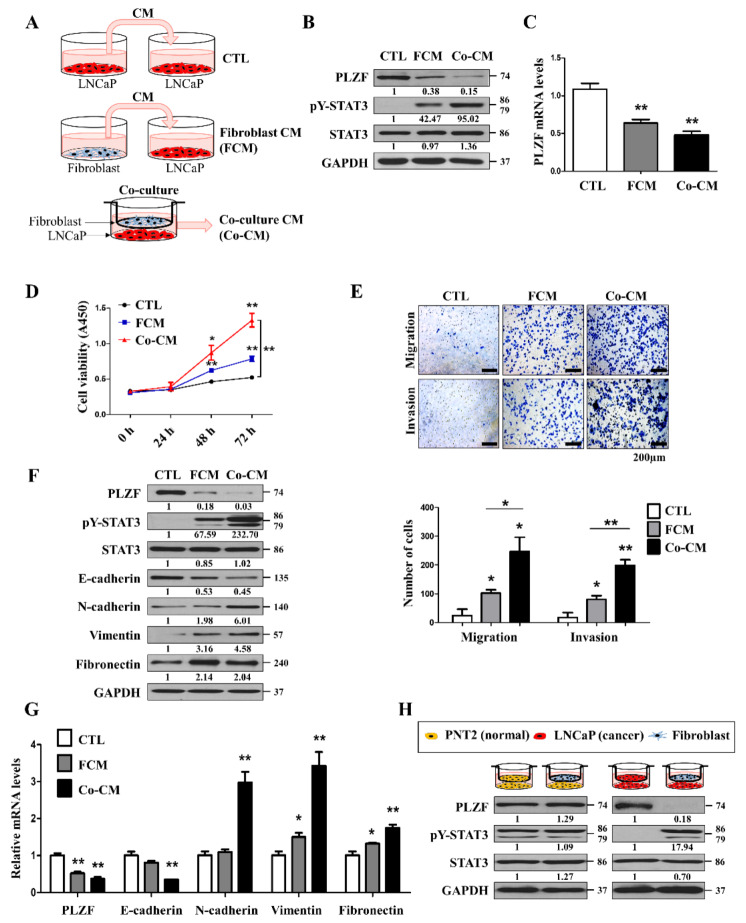
PLZF is reduced in prostate cancer due to the prostate tumor-associated fibroblasts. (**A**) Schematic of the experimental set-up for conditioned media (CMs) treatment and the Transwell co-culture system. Conditioned media (CM) were collected from fibroblast (FCM) that had been incubated for 48 h. LNCaP cells were cultured in a mixture of conditioned media and fresh media for 24 h (=FCM). LNCaP cells were co-cultured with fibroblast using a cell culture chamber for 24 h and the cells in the lower chamber were lysed. (**B**) LNCaP cells with FCM or co-cultured with fibroblast for 24 h. LNCaP cells were lysed for Western blotting. (**C**) mRNA expression levels of PLZF were examined by qRT-PCR in LNCaP cells with FCM or co-culture. (**D**) Cell growth rates between FCM-treated cells and co-culture CM (Co-CM) were compared using a CCK assay. (**E**) Transwell migration and Matrigel invasion assays were conducted in LNCaP cells with FCM or Co-CM. The relative cell numbers are shown (bottom). Representative images from three independent experiments were quantified as the mean ± SD. (**F**) Protein expression levels of EMT markers in LNCaP cells with FCM or co-culture. (**G**) mRNA expression levels of epithelial–mesenchymal-transition (EMT) markers were examined by qRT-PCR in LNCaP cells with FCM or co-culture. (**H**) Prostate normal cells (PNT2) and prostate cancer cells (LNCaP) were co-cultured with fibroblast for 24 h. PLZF, pY-STAT3, STAT3, and GAPDH protein expression by Western blotting in LNCaP cells. * *p* < 0.05, ** *p* < 0.01. The uncropped blots and molecular weight markers of Figure 5 are shown in Appendix A

**Figure 6 cancers-12-01083-f006:**
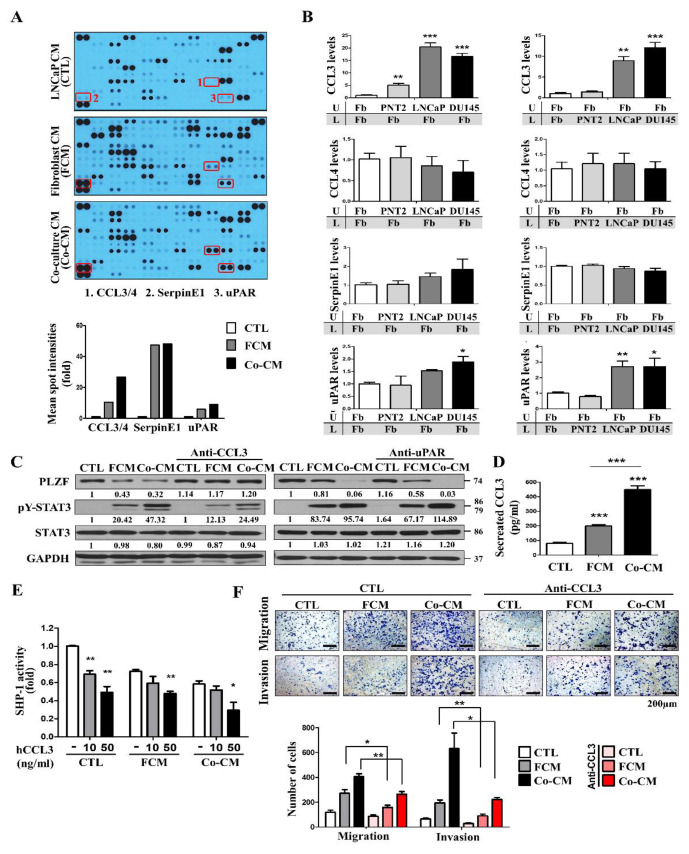
Loss of CCL3 inhibits the fibroblast-induced prostate cancer cells migration and invasion. (**A**) Each CM was applied to the cytokine arrays (top). The cytokines in boxes are more enriched in co-cultured CM than in fibroblast CM. Mean intensities of cytokines are plotted (bottom). (**B**) PNT2, LNCaP, and DU145 cells were co-cultured with fibroblast in a cell culture chamber for 24 h. Cells in a lower chamber were lysed for qRT-PCR. (**C**) LNCaP cells were treated with the indicated conditioned media with the addition of an anti-CCL3 antibody or anti-uPAR antibody and then subjected to Western blotting. (**D**) Concentrations of CCL3 in the indicated conditioned media were analyzed using an ELISA kit. Summarized results from three independent experiments were quantified as the mean ± SD. (**E**) Cells, which had been transfected with the luciferase plasmid, were pre-treated with recombinant protein CCL3 for 4 h, incubated under FCM or Co-CM for 24 h. (**F**) LNCaP cells, which had been pre-treated with anti-CCL3 antibody for 4 h, were incubated in the indicated CMs for 24 h. The relative cell numbers are shown (right). In B, D, E, and F, representative images from three independent experiments were quantified as the mean ± SD; * *p* < 0.05, ** *p* < 0.01, *** *p* < 0.001. The uncropped blots and molecular weight markers of Figure 6 are shown in Appendix A

**Figure 7 cancers-12-01083-f007:**
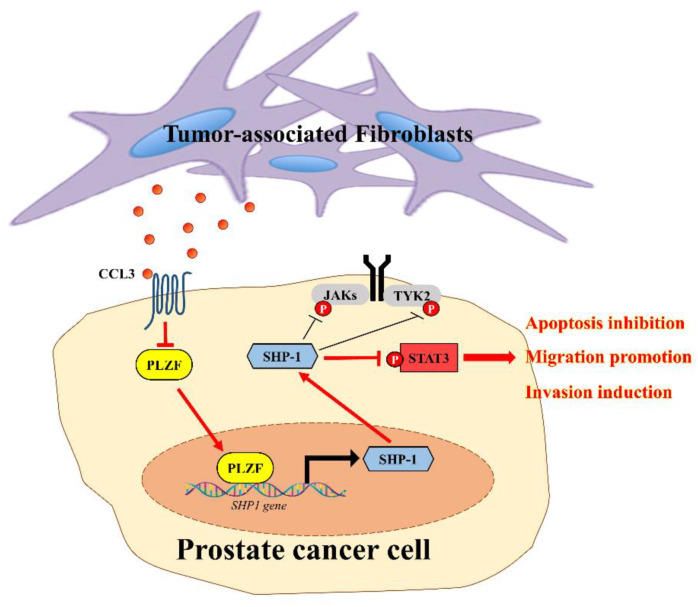
A schematic illustration of the tumor suppressor PLZF actions in prostate cancer progression.

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
