# Peer review of "Crosstalk between Prostate Cancer Cells and Tumor-Associated Fibroblasts Enhances the Malignancy by Inhibiting the Tumor Suppressor PLZF"

_cancers, 2020, doi:10.3390/cancers12051083_

Round 1

Reviewer 1 Report

I accept the manuscript in the present form.

Reviewer 2 Report

Accept this present form

Reviewer 3 Report

In the resubmitted manuscript " Crosstalk between prostate cancer cells and tumor 2 associated fibroblasts enhances the malignancy by 3 inhibiting tumor suppressor PLZF" the authors have complied with most of the suggestions provided in the initial submission. This has enhanced the quality of the manuscript and may be accepted for publication.

This manuscript is a resubmission of an earlier submission. The following is a list of the peer review reports and author responses from that submission.

Round 1

Reviewer 1 Report

This interesting study is about prostate cancer cells and tumor associated fibroblasts enhances the malignancy by inhibiting tumor suppressor PLZF. The authors found that PLZF has putative tumor suppressor function in prostate cancer by inhibiting phosphorylation of STAT3. Compared with the benign prostate, high-grade prostate cancer patient tissue was negatively correlated with PLZF expressions. PLZF depletion accelerated proliferation and survival, migration, and invasion in human prostate cancer cells. It’s an interesting study with robust research data proven their hypotheses. There are some concerns in this study and need the authors to address.

Major Concerns:

If possible , please add cell-line data with PC-3 cell line to make your study more convincible. In Figure 5, very impressive results of PLZF is reduced in prostate cancer due to the prostate tumor-associated fibroblasts in LNCaP cell line , do these results are the same in DU-145 cell line? In Figure 7, your schematic illustration of the tumor suppressor PLZF actions in prostate cancer progression. You need to do detail describe about reduction of SHP-1 induce blockage of SHP-2 and what reaction happen then such as reduce proliferation? or reduce migration? And please draw more detail about your schematic illustration. I still doubt that PLZF multiple function to reduce prostate cancer and to tumor-associated fibroblasts. You need to offer more evidences not simple loss of CCL3 to inhibit tumor migration. You summarize high-grade prostate cancer patient tissue was negatively correlated with PLZF expressions. I wonder that are there any difference PLZF expressions in CSPC and CPRC? For CRPC status had very poor prognosis in prostate cancer patients, if PLZF play a role in CRPC may be a useful potential therapeutic target. Please explain.

Minor comments:

The authors need to edit this paper for professionally English language.

Author Response

Please see the attachment for response to the reviewer’s comments.

Reviewer 2 Report

*. Please explain why you choose the PLZF.

*. Apoptosis experiment should be done.

*. Cell invasive ability should be added cell tracking experiment.

*. English editing should be done. 

Author Response

(The authors gave the same response as above.)

Reviewer 3 Report

The manuscript by noh et al is well represented and would add value to the current set of info. The manuscript may need additional information.

Although it has been suggested that Ser727 phosphorylation is a secondary event after Tyr705 phosphorylation required for the maximal transcriptional activity of STAT3. Therefore the authors should address the status of Ser727 and if Stat3 is translocated into the nucleus. The authors have to address whether if there is an inhibition of Stat3 into the nucleus as a result of PLZF activation?

Figure 3A: The authors need to determine the % of Closure in scratch Assay.

In the methodology, the authors will have to provide the methods for cell viability assay and transfection with siRNA or overexpression.

Author Response

(The authors gave the same response as above.)
